# Chalcone Scaffolds Exhibiting Acetylcholinesterase Enzyme Inhibition: Mechanistic and Computational Investigations

**DOI:** 10.3390/molecules27103181

**Published:** 2022-05-16

**Authors:** Yossra A. Malik, Talal Ahmed Awad, Mohnad Abdalla, Sakina Yagi, Hassan A. Alhazmi, Waquar Ahsan, Mohammed Albratty, Asim Najmi, Shabbir Muhammad, Asaad Khalid

**Affiliations:** 1Department of Biochemistry, Medicinal and Aromatic Plants and Traditional Medicine Research Institute, National Center for Research, Khartoum 11111, Sudan; yossraosman@gmail.com (Y.A.M.); talaladlan@hotmail.com (T.A.A.); 2Unit of Molecular Genetics, Central Laboratory, Khartoum 11111, Sudan; 3Department of Pharmaceutical Chemistry, Faculty of Pharmacy, Ibn Sina University, Khartoum 10995, Sudan; 4Key Laboratory of Chemical Biology (Ministry of Education), Department of Pharmaceutics, School of Pharmaceutical Sciences, Cheeloo College of Medicine, Shandong University, 44 Cultural West Road, Jinan 250012, China; mohnadabdalla200@gmail.com; 5Department of Botany, Faculty of Science, University of Khartoum, Khartoum 11115, Sudan; sakinayagi@gmail.com; 6Substance Abuse and Toxicology Research Center, Jazan University, P.O. Box 114, Jazan 45142, Saudi Arabia; haalhazmi@jazanu.edu.sa; 7Department of Pharmaceutical Chemistry, College of Pharmacy, Jazan University, P.O. Box 114, Jazan 45142, Saudi Arabia; wmohammad@jazanu.edu.sa (W.A.); malbratty@jazanu.edu.sa (M.A.); anajmi@jazanu.edu.sa (A.N.); 8Department of Chemistry, College of Science, King Khalid University, P.O. Box 9004, Abha 61413, Saudi Arabia; mshabbir@kku.edu.sa

**Keywords:** acetylcholinesterase inhibitors, chalcones, molecular docking, mechanistic, molecular dynamics simulations, in silico

## Abstract

This study was aimed to perform the mechanistic investigations of chalcone scaffold as inhibitors of acetylcholinesterase (AChE) enzyme using molecular docking and molecular dynamics simulation tools. Basic chalcones (**C1**–**C5**) were synthesized and their in vitro AChE inhibition was tested. Binding interactions were studied using AutoDock and Surflex-Dock programs, whereas the molecular dynamics simulation studies were performed to check the stability of the ligand–protein complex. Good AChE inhibition (IC_50_ = 22 ± 2.8 to 37.6 ± 0.75 μM) in correlation with the in silico results (binding energies = −8.55 to −8.14 Kcal/mol) were obtained. The mechanistic studies showed that all of the functionalities present in the chalcone scaffold were involved in binding with the amino acid residues at the binding site through hydrogen bonding, π–π, π–cation, π–sigma, and hydrophobic interactions. Molecular dynamics simulation studies showed the formation of stable complex between the AChE enzyme and **C4** ligand.

## 1. Introduction

Acetylcholinesterase (AChE) enzyme belongs to the acetylhydrolase family of enzymes responsible for the hydrolysis of choline esters, such as acetylcholine in the synapse. Acetylcholine (ACh) is the primary neurotransmitter of the parasympathetic nervous system, which plays an essential role in arousal, memory, and attention. Inhibition of the AChE enzyme increases the concentration of ACh in the synapse. In addition, this approach is currently utilized in the treatment of cognitive disorders, such as Alzheimer’s disease (AD). AD is characterized by the decline and deterioration of memory and cognitive skills and is the most common cause of dementia contributing to 60–80% of all cases. The use of AChE inhibitors has been accepted as one the most effective treatment strategies against AD [1,2] and several AChE inhibitors acting as cognitive enhancers are currently being investigated for this purpose. To date, Donepezil, Rivastigmine, and Galantamine have received approval from the United States Food and Drug Administration (USFDA) [3,4]. However, mild to moderate side effects, including nausea, vomiting, diarrhea, syncope, and bradycardia are often experienced with these medications, which beset their use. Therefore, an urgent need exists for the development of newer, safer, and more effective AChE inhibitors.

AChE enzyme has a large but flexible structure consisting of 535 amino acids with a 20 Å deep and 5 Å wide narrow tunnel leading up to its active site [5]. The tunnel, also known as the active-site gorge, consists of 14 flexible aromatic amino acid residues [6]. These aromatic residues play essential roles in binding to the substrate. For instance, W84 and F330 are responsible for the π–cation interaction between the amino acid and the quaternary ammonium cation of the choline substrate [7]. Three amino acids are essential for the AChE activity and are involved in the transfer of acetyl group, including Ser200, His440, and Glu327, forming a catalytic triad. The active site of the AChE enzyme has two subunits: The catalytic “esteric” subunit and the “anionic” subunit with a chlorine binding pocket [8]. The diameter of the quaternary ammonium group of the choline substrate is considerably larger in size than the cross-section present at the narrowest point of the anionic subunit. Therefore, AChE must make significant movements and adjustments to activate its catalytic function [9,10]. The anionic site of the AChE enzyme consists of amino acids Trp84, Tyr130, Phe330, and Phe331, which facilitate the binding of quaternary ammonium cation to the enzyme. Several in vitro and in silico approaches have been carried out to explore the recognition pattern between the AChE enzyme and its inhibitors at the molecular level [11,12,13,14]. AChE from Torpedo californica is a common and well-studied source of AChE that has been utilized to study AChE inhibitory actions. [15]. The active site of Torpedo AChE is remarkably identical to the mouse and human AChE. The sole small variation is the mutation in Tyr337 in human to Phe330 in Torpedo, which has no significant effects on the electrostatic or steric characteristics of the active site [16].

Chalcones (1,3-diaryl-2-propen-1-ones) are one of the most important synthetic and natural scaffolds. Naturally, they are present in fruits, vegetables, spices, tea, and soy-based foodstuffs and act as the precursors of flavonoids and isoflavonoids [17]. Structurally, they are open chain flavonoids in which the two aromatic rings (A and B) are joined together by a three-carbon α, β-unsaturated carbonyl system (Figure 1). The aromatic ring to which the carbonyl group is attached is designated as ring A, while the other aryl group is called ring B. The α, β-unsaturated carbonyl functionality of chalcones acts as Michael acceptor, which interacts with thiols and the sulfhydryl group of cysteine residues present at the binding site. This interaction is believed to be responsible for the versatile biological properties of chalcones [18,19].

Previously, several chalcone derivatives were synthesized and tested against the AChE enzyme in vitro [20,21,22,23] and the promising results demonstrated the importance of the chalcone scaffold in binding and inhibiting the enzyme. Chalcones and their synthetic analogs have shown potential activities against various neurological diseases, particularly for AD [24,25,26,27,28]. The modifications made in the chalcone scaffold were mainly directed to the substitution in the aromatic rings. These substitutions include the attachment of heteroaryl groups of varying size to the phenyl rings A and B, or substitution using electron-donating and releasing substituents of varying polarity, or a combination of both. These studies motivated us to synthesize and test the AChE inhibitory activities of common chalcone scaffolds in vitro, as well as examine their mechanism of action in silico using molecular docking and molecular dynamics simulation tools. Mechanistic investigations of the binding of chalcones to the AChE receptor using different modeling techniques would help in designing newer and better chalcone analogs that could show improved binding to the AChE receptor.

## 2. Materials and Methods

### 2.1. Chemicals and Instruments

All of the solvents and reagents used for synthesis were of laboratory reagent (LR) grade and were purchased from Sigma-Aldrich, Steinheim, Germany. The completion of reaction and purity was monitored by thin-layer chromatography (TLC) using Merck precoated silica plates (G F254). The solvent system used for performing TLC was a mixture of petroleum ether/ethyl acetate (8:2). The proton nuclear magnetic resonance (^1^H NMR) spectra were recorded on a Bruker Avance III 500 MHz instrument (Billerica, MA, USA) using dimethylsulfoxide (DMSO) as solvent. Chemical shifts were reported in parts per million (ppm) down the field using tetramethylsilane (TMS) as the internal standard. Mass spectra were obtained using an electrospray ionization (ESI) probe as an ion source on Finnigan LTQ HPLC-MS spectrophotometer (Thermo Fischer Scientific, CA, USA). Elemental analyses were performed on Thermo Quest CE Elemental analyzer (Thermo Fisher Scientific, Carlsbad, CA, USA).

### 2.2. General Procedure for the Synthesis of Compounds ***C1***–***C5***

The titled compounds (**C1**–**C5**) were synthesized via Claisen–Schmidt condensation, according to the scheme given in Figure 2 and using the standard procedure described elsewhere with slight modification [29]. An equimolar mixture of substituted acetophenones and substituted benzaldehydes in ethanol (20 mL) was stirred on an ice bath, followed by the addition of 10% sodium hydroxide solution (2.5 mL) dropwise. Continuous stirring was applied for 30 min to 1 h and the reaction completion was monitored using TLC. The resulting mixture remained overnight in the refrigerator and the separated solid was filtered, washed with ice-cold water, then followed by cold ethanol. The final product was recrystallized using ethanol. For the preparation of hydroxylated chalcones, the amount of sodium hydroxide solution was doubled, and continuous stirring was applied for 24–48 h, followed by quenching with 1% aqueous hydrochloric acid. 


(*E*)-1-(4-chlorophenyl)-3-phenylprop-2-en-1-one (**C1**)Yield 95%. ^1^H NMR (500 MHz, CDCl_3_) δ 7.94 (d, *J* = 5.4 Hz, 2H), 7.79 (d, *J* = 5.7 Hz, 1H), 7.62 (d, *J* = 3.1 Hz, 2H), 7.46 (dd, *J* = 3.2, 3.5 Hz, 3H), 7.42–7.37 (m, 3H). MS (*m/z*) calculated for C_15_H_11_ClO 242.05 found (M^+^) 243.11 (100%); Analysis calc. C, 74.23; H, 4.57; found C, 74.23; H, 4.49.(*E*)-1-(4-ethylphenyl)-3-(4-nitrophenyl)prop-2-en-1-one (**C2**)Yield 77%. ^1^H NMR (500 MHz, DMSO) δ 8.30 (d, *J* = 5.8 Hz, 2H), 8.20 (s, 1H), 8.18 (d, *J* = 4.9 Hz, 1H), 8.14 (dd, *J* = 2.3, 2.8 Hz, 3H), 7.82 (d, *J* = 5.7 Hz, 1H), 7.44 (d, *J* = 4.1 Hz, 2H), 2.73 (q, *J* = 3.6 Hz, 2H), 1.23 (t, *J* = 4.6 Hz, 3H). MS (*m/z*) calculated for C_17_H_15_NO_3_ 281.11 found (M^+^) 282.15 (100%); Analysis calc. C, 72.58; H, 5.37; N, 4.98; found C, 72.33; H, 5.21; N, 4.99.(*E*)-1-(4-chlorophenyl)-3-(4-methoxyphenyl)prop-2-en-1-one (**C3**)Yield 78%. ^1^H NMR (500 MHz, CDCl_3_) δ 7.90 (t, *J* = 4.8 Hz, 2H), 7.76 (d, *J* = 4.6 Hz, 1H), 7.58 (d, *J* = 4.7 Hz, 2H), 7.44 (d, *J* = 3.5 Hz, 2H), 7.33 (d, *J* = 4.6 Hz, 1H), 6.92 (d, *J* = 3.7 Hz, 2H), 3.84 (s, 3H). MS (*m/z*) calculated for C_16_H_13_ClO_2_ 272.06 found (M^+^) 273.17 (100%); Analysis calc. C, 70.46; H, 4.80; found C, 70.61; H, 4.72.(*E*)-1-(4-bromophenyl)-3-(4-hydroxyphenyl)prop-2-en-1-one
(**C4**)Yield 59%. ^1^H NMR (500 MHz, DMSO) δ 10.15 (s, 1H), 8.08 (d, *J* = 3.5 Hz, 2H), 7.77 (t, *J* = 3.8 Hz, 4H), 7.72 (s, 2H), 6.85 (d, *J* = 4.5 Hz, 2H). MS (*m/z*) calculated for C_15_H_11_BrO_2_ 301.99 found (M^+^) 301.24 (100%); Analysis calc. C, 59.43; H, 3.66; found C, 59.33; H, 4.08.(*E*)-1-(4-bromophenyl)-3-(4-nitrophenyl)prop-2-en-1-one
(**C5**)Yield 83%. ^1^H NMR (500 MHz, DMSO) δ 8.30 (d, *J* = 3.8 Hz, 2H), 8.19 (d, *J* = 3.7 Hz, 2H), 8.16–8.11 (m, 3H), 7.84 (t, *J* = 4.0 Hz, 3H). MS (*m/z*) calculated for C_15_H_10_BrNO_3_ 330.98 found (M^+^) 331.21 (100%); Analysis calc. C, 54.24; H, 3.03; N, 4.22; found C, 54.23; H, 2.89; N, 4.14.


### 2.3. In Vitro AChE Inhibition Activity

#### 2.3.1. AChE Inhibition Assay

AChE enzyme from electric eel (*Ee*AChE) (type VI), acetylthiocholine iodide, Ellman reagent, 5,5-dithiobis-2-nitro benzoic acid (DTNB), and Eserine (standard inhibitor) were purchased from Sigma-Aldrich, Steinheim, Germany. AChE inhibitory activity was performed in vitro using a modified spectrophotometric method developed by Ellman et al. [30]. Experiments were carried out in a 96-well microtiter plate using a Multiskan spectrum spectrophotometer. The test compounds were prepared immediately before the experiment as fresh samples. Each compound was dissolved in 200 μL methanol, 150 μL 0.1 mM phosphate buffer (pH 8.0), and 20 μL AChE enzyme. An aliquot of 10 μL of resulting test solutions was mixed and incubated at 25 °C for 10 min. The sample with 10 μL of solvent (methanol) rather than the test compound served as a negative control. After 10 min, 10 μL DTNB was added to the wells and the reaction was initiated with the addition of 10 μL acetylthiocholine (ATCh). All of the samples were prepared and analyzed in triplicate. The hydrolysis of ATCh was monitored at wavelength 412 nm by measuring the formation of a yellow anion of 5-thio-2-nitrobenzoate due to the reaction of DTNB with thiocholine. The initial rate was measured as the change in optical density/min (OD/min). As the extinct coefficient of the yellow anion is known, the following equation was used to calculate the rate of the enzymatic reaction: (1)$$\mathrm{Rate}\;(\mathrm{moles}/L/\min)\;=\;\frac{\mathrm{change}\;\;\mathrm{in}\;\;\mathrm{absorbance}}{\varepsilon_{TNB}}\;$$
where *ε_TNB_* is the molar extinction coefficient value of 5-thio-2-nitrobenzoate = 13.6 × 10^3^ at 412 nm [31].

The following formula was used to calculate the % enzyme inhibition by test compounds:(2)$$\%\;\;\mathrm{inhibition}\;=\;1-\frac{\mathrm{Rate}\;\;\mathrm{of}\;\mathrm{change}\;\;\mathrm{in}\;\;\mathrm{the}\;\;\mathrm{absorbance}\;\;\;\mathrm{of}\;\;\mathrm{test}}{\mathrm{Rate}\;\;\mathrm{of}\;\mathrm{change}\;\;\mathrm{in}\;\;\mathrm{the}\;\;\mathrm{absorbance}\;\;\;\mathrm{of}\;\;\mathrm{cotrol}}\;\times \;100$$

#### 2.3.2. IC_50_ Determination

Chalcones that showed ~50% and more inhibition activity were considered active, and their IC_50_ values were determined. Chalcone solutions were serially diluted to different concentrations (0.0977–50 µM) and were analyzed in triplicate using the above-described modified Ellman’s method. Finally, IC_50_ values were determined and statistically analyzed using EZ-Fit Enzyme Kinetic Program (Perrella Scientific Inc., Amherst, MA, USA).

### 2.4. In silico Mechanistic Molecular Docking Studies

The three-dimensional (3D) structures of all the synthesized compounds were drawn, and energy was minimized using Chem3D software and saved in *.mol2 format [32]. The 3D structure of *Torpedo californica Ee*AChE was retrieved from the protein data bank (PDB) website as *N*-piperidinopropyl-galanthamine complex (PDB ID: 3I6M) with the resolution of 2.19 Å [33]. The 3D structures of chalcones were docked against the protein using two docking tools: The academic docking tool AutoDock 4.2.2 [34] and the commercial docking tool Surflex-Dock 2.1 [35] for comparison purposes. Ligands were prepared by adding all hydrogens and by computing Gasteiger partial atomic charges and the final ligand structures were saved in *.pdbqt format. Protein was prepared by adding all polar hydrogens and by calculating the partial Kollman charges for an arbitrary molecule. The final protein structure was also saved in the *.pdbqt format. The grid box was created using the auto grid module and its volume was set to the maximum value (126 × 126 × 126 Å) to allow the ligand to move freely. The above file was saved as *.gpf format followed by the creation of a docking parameter file, which was saved in *.dpf format. The Lamarckian genetic (LG) algorithm was used to generate poses and to calculate the binding energies for each test compound. All the docking results were visualized and analyzed using Discovery Studio Visualizer (DSV; Accelrys Discovery Studio 2.5) and Ligplot 4.5.3 software.

### 2.5. Molecular Dynamics (MD) Simulation Studies

To mimic the aqueous environment around the protein, MD simulation studies were also performed for the AChE enzyme and its complex with **C4** ligand. The MD simulations were performed for a time of 60 ns (6 × 10^6^ femtosecond) at a temperature of 310 K. Energy minimization was carried out for 10,000 steps. Equilibration was performed for a period of 1 ns at normal pressure and temperature. At a later time, a production run was carried out for 60 ns at constant volume and temperature (NVT). All of the calculations of MD simulations were performed using the CHARMM force-field [36] in NAMD [37]. The VMD program was used to calculate the root mean square deviation (RMSD) and root mean square fluctuation (RMSF) [38]. The MD simulations input parameters remained as described in our previous studies [39,40]. The RMSD and RMSF values were analyzed for the assessment of conformational stability and residual flexibility of free AChE enzyme and its **C4** complex.

## 3. Results and Discussion

### 3.1. Characterization of Titled Compounds (***C1***–***C5***)

The characterization of all the synthesized compounds (**C1**–**C5**) was performed using ^1^H NMR and mass spectroscopic techniques. The CHNS analyzer confirmed the purity of all compounds, and the values obtained were within limits. The ^1^H NMR and mass spectra of all the titled compounds are provided in the Appendix A

### 3.2. In Vitro AChE Inhibitory Activity

The AChE inhibitory activity of the synthesized chalcones was experimentally tested in vitro against the AChE enzyme. Results showed moderate to good inhibitory activity of all compounds with IC_50_ values in the range of 22 ± 2.8–37.6 ± 0.74 μM (Table 1). The CLogP values of all compounds were calculated using ChemDraw Ultra 6.0 software and were found to be in the range of 4.13–4.65.

### 3.3. In Vitro−In Silico Correlation Studies and Mechanistic Investigations

The in vitro−in silico correlation studies were performed to investigate the correlation between the in vitro AChE inhibitory activities of the compounds with their in silico molecular docking results. Two commonly employed docking tools AutoDock and Surflex-Dock were used to calculate the binding energies and to determine the type of interactions between the synthesized chalcones and the AChE receptor.

Compound **C1** exhibited the highest AChE inhibition activity among all the tested compounds, with an IC_50_ value of 22 ± 2.8 μM. When subjected to molecular docking, the results obtained using AutoDock software showed binding energy of −8.4 Kcal/mol. Two hydrogen bonds were predicted between the O16 atom (carbonyl group) of compound **C1** and N atom of His440 and γ-Oxygen atom (OG) atom of Ser200 at the active site with distances of 3.1 and 3.0 Ǻ, respectively. One π–π interaction was also monitored between ring B and Phe330 with 3.8 Ǻ. Surflex-Dock software showed the formation of only one hydrogen bond between the Cl atom of compound **C1** and OH of Tyr130 at a distance of 2.7 Ǻ. Additionally, four π–π interactions were monitored, two of them between ring B of compound **C1** and Phe330, Tyr334 residues at distances of 3.6 and 4.9 Ǻ, respectively. Ring A of the chalcone also showed the formation of two π–π interactions with Trp84 at distances of 4.5 and 3.7 Ǻ. The AutoDock tool showed hydrophobic interactions between the compound **C1** and Gly117, Tyr130, Gly123, Gly118, Phe330, Trp84, Ile439, and Tyr442 residues, while Surflex-Dock showed hydrophobic interactions between the compound and Tyr334, Gly117, Trp84, and Phe330 residues. The interactions of compound **C1** with the AChE enzyme are shown in Figure 3.

Compound **C2** also showed good AChE inhibitory activity with an IC_50_ value of 31.2 ± 1.8 μM. The AutoDock software showed binding energy of −8.29 Kcal/mol and predicted four hydrogen bonds, two of them were present between the O20 atom (nitro group) of compound **C2** and ε-Oxygen atom-1 (OE1) atom of Gln69 and O atom of Tyr70 at a distance of 2.3 Ǻ. The other two hydrogen bonds were predicted between the O21 atom (nitro group) of compound **C2** and O atom of Tyr70 and δ-Oxygen atom-1 (OD1) atom of Asn85 at distances of 2.8 and 2.7 Ǻ, respectively. In addition, two π–π interactions were also predicted between ring A and Trp84 residue present at the active site at a distance of 3.9 Ǻ, and the π–cation interaction between ring A and NE1 of His440 at a distance of 5.1 Ǻ. These multiple interactions helped in increasing the binding affinity of the compound. Surflex-Dock also showed the presence of four hydrogen bonds; two between the α-Oxygen atom (OA) atom (nitro group) of compound **C2** and OE1 of Glu199 at distances of 2.8 and 2.9 Ǻ, and two between the O atom (carbonyl group) of compound **C2** and NE2 of His440 and O atom of Ser200 at distances of 10.5 and 11.3 Ǻ, respectively. Hydrophobic interactions were also predicted between the compound **C2** and the His440, Gly441, Tyr442, Ile430, Phe330, Trp84, Gly118, and Ser122 amino acid residues. Surflex-Dock also showed hydrophobic interactions with Tyr334, Tyr121, Phe330, Asp72, and Gly118 residues. Binding interactions between compound **C2** and the surrounding amino acid residues present at the active sites are depicted in Figure 4.

Similarly, the calculated binding energy of compound **C3** using the AutoDock program was found to be −8.14 Kcal/mol. Moderate interactions at the binding site were predicted, which were in correlation with its moderate IC_50_ values of 32 ± 2.4 µM against the AChE enzyme. The AutoDock tool predicted two hydrogen bonds between the O18 atom (methoxy group) of compound **C3** and two N-atoms of Gly123 at distances of 3.0 and 2.8 Ǻ. One π–π interaction between ring A and Phe330 at a distance of 3.4 Ǻ and the π-sigma interaction between ring B and Trp84 of active site were monitored. On the other hand, the Surflex-Dock tool predicted five hydrogen bonds; three of them between the carbonyl O atom of compound **C3** and OG atom of Ser122, OH atom of Tyr442, and NE1 atom of Trp432 at distances of 2.9, 2.7, and 2.9 Ǻ, respectively. Two others were present between the Cl atom of compound **C3** and OG atoms of Ser124 and Gln69 residues at distances of 3.0 and 3.3 Ǻ, respectively. Two π–π interactions were monitored between ring B and Trp84 at distances of 4.0 and 5.7 Ǻ. Hydrophobic interactions predicted by the AutoDock tool were between compound **C3** and the Trp84, His440, Tyr442, Tyr432, Phe330, Ile439, Gly123, Pro56, Asn85, Ser124, Ser122, and Gln69 amino acid residues at the active site, whereas, the Surflex-Dock showed hydrophobic interactions with Gly123, Trp84, Leu127, Ser122, Phe330, Tyr334, Trp432, and Met436 residues. Binding interactions between compound **C3** and the amino acid residues present at the active site of AChE are presented in Figure 5.

Compound **C4** also showed good binding affinity value (−8.55 Kcal/mol) with the AChE receptor using the AutoDock tool in correlation with the in vitro results (IC_50_ = 36.9 ± 5.6 µM). Two hydrogen bonds were predicted between the O17 atom (hydroxyl group) of compound **C4** and OE1 atom of Gln69 and OG atom of Ser124 at distances of 2.5 and 2.8 Ǻ, respectively. Ring A participated in three π–π interactions, one with Phe330 and two others with Trp84 residue at distances of 4.0, 3.8, and 4.3 Ǻ, respectively. Surflex-Dock also predicted the formation of two hydrogen bonds between the O17 atom (hydroxyl group) of compound **C4** and Glu199 and Tyr130 residues at a distance of 2.6 Ǻ; two π–π interactions between ring B of compound **C4** and Trp84 at distances of 3.6 and 4.8 Ǻ; two π–π interactions between ring A and Trp84 and Phe330 residues at distances of 4.5 and 3.4 Ǻ. The hydrophobic interactions predicted by AutoDock at the active site were observed with Gly123, Ser122, Phe330, and Trp84 residues, while Surflex-Dock predicted hydrophobic interactions with Trp432, Tyr334, Phe330, Trp84, Glu199, Ile444, and Gly117 residues. Binding interactions between compound **C4** and the amino acid residues present at the AChE active site are presented in Figure 6.

Finally, compound **C5** showed binding energy of −8.16 Kcal/mol in the AutoDock and a considerable IC_50_ value of 37.6 ± 0.75 µM in the in vitro inhibition assay, which was the least among all the tested compounds. The AutoDock tool showed the formation of five hydrogen bonds. The O20 atom (nitro group) of compound **C5** formed three H-bonds with the O atom of Tyr70, N atom of Asp72, and OD1 atom of Asn85 at distances of 3.1, 3.0, and 2.4 Ǻ, respectively. The hydrogen bonds were also predicted between the O19 atom (nitro group) of compound **C5** and OE1 of Gln69 at a distance of 2.5 Ǻ and O2 of Tyr70 at a distance of 2.3 Ǻ. The AutoDock software also monitored one π–cation interaction between ring A and NE2 of His440 at a distance of 6.1 Ǻ, and two other interactions between ring A and Trp84 at distances of 4.1 and 3.5 Ǻ. On the other hand, Surflex-Dock software also showed the presence of five hydrogen bonds; one between the O20 atom (nitro group) of compound **C5** and OE1 atom of Glu199 at a distance of 2.8 Ǻ, and four other bonds between the O19 atom (nitro group) of the compound and OE1 atom of Glu199 at distances of 3.1 and 2.9 Ǻ, NE2 atom of His440 at a distance of 3 Ǻ, and OG atom of Ser200 at a distance of 3 Ǻ. No π interactions were predicted for this compound using Surflex-Dock, which could explain the lesser binding energy in correlation with the higher IC_50_ values. Hydrophobic interactions predicted by the AutoDock were between the compound and Ser122, Tyr442, Trp84, and Phe330 residues at the active site. On the other hand, Surflex-Dock showed hydrophobic interactions with Asp72, Phe330, Tyr121, Tyr334, Tyr70, and Gly118 residues. Binding interactions between compound **C5** and the amino acid residues present at the active site are presented in Figure 7.

Chalcones have been regarded as a privileged scaffold due to their extraordinary features of smaller molecular size, convenient and cost-effective synthesis, and flexibility for modifications [41,42]. The lipophilicity of the corresponding compounds can be altered and modulated to allow it to cross the blood–brain barrier (BBB) and be effective for neurological disorders, including AD [43,44,45]. The critical capabilities of chalcone scaffold led to vast investigations resulting in the identification of several derivatives showing promising anticancer, anti-infection, anti-ulcer, anti-inflammatory, and anti-neurodegenerative diseases [46,47,48,49]. Hybridization of chalcones with heterocyclic rings, such as pyrazole and indole, yielded compounds with good antiproliferative activities [50]. In one study, various pyrazolic chalcone derivatives were synthesized and tested against Huh7, MCF7, and HCT116 cell lines. Four out of 42 tested compounds showed promising antiproliferative activities with IC_50_ values in the range of 0.5–4.8 µM [51]. The capacity of chalcones and their heteroanalogues to behave as activated unsaturated systems in carbanions conjugate addition processes in the presence of base catalysts is one of their most important characteristics. Chalcones have been utilized as popular substrates for the generation of a variety of heterocyclic, carbocyclic, and flavonoid derivatives. Flavonoids containing chalcone functionality are the hydroxylated phenolic compounds, which are known to possess important biological activities [52,53].

The flexibility of chalcone scaffold and the presence of critical functional groups in its structure allow it to bind to various molecular targets and show different biological effects. One of the most widely explored targets for chalcones is the AChE enzyme and several previously conducted studies reported various chalcone derivatives showing the AChE inhibiting activity. In most of the studies, chalcones were utilized as a basic skeleton and various structural analogs were prepared by modifying the two lipophilic aryl rings A and B. In this way, the lipophilicity of resulting compounds was optimized to allow it to cross the BBB and bind to the AChE receptor. The promising results obtained in those studies prompted us to carry out the mechanistic studies on chalcone scaffold and study the effects and roles of different functionalities present in chalcone in binding to the AChE enzyme. In this study, five substituted diphenylchalcones were prepared using a previously reported procedure, and their in vitro AChE inhibition assay was performed using the standard method. All of the synthesized chalcones displayed moderate to good inhibitory activity and the modification in the substituent on the aromatic ring showed an interesting effect on the enzyme inhibition. 

In vitro enzyme inhibition assay results were correlated with the in silico molecular docking results, where the structures of synthesized compounds were docked in the AChE protein using two widely employed docking tools: AutoDock (academic) and Surflex-Dock (commercial). Two docking tools were selected to compare their results and to identify all of the binding interactions that may be present between the chalcone scaffold and the amino acid residues present at the active-site gorge of the protein. Interestingly, both the docking tools were able to identify and predict many interactions between the functionalities present in chalcone and the amino acids at the binding site (Figure 8).

The significant functionalities present in the chalcone scaffold are the carbonyl group (-C=O), the α, β-unsaturation (double bond), aryl rings A and B, and the substitutions on the aryl ring (R and R_1_). Using molecular docking studies, the molecular interactions presented by these groups were predicted; their binding energies (docking score) were calculated and correlated with the in vitro enzyme inhibition results. Carbonyl functionality was predicted to be involved in forming hydrogen bonds with the Histidine and Serine residues present at the binding site of the protein. The aryl rings A and B showed the π–π interaction with the aromatic amino acids Phenylalanine, Tryptophan, and Tyrosine. These aryl rings were also shown to be involved in the π–cation interaction between the aryl ring and the quaternary nitrogen atom (ammonium cation) present in amino acids, such as Histidine. Substitutions on the aromatic ring also had considerable effects on the binding affinity of compounds. The Cl atom present in compound **C1** was involved in the interaction with the –OH group present in Tyrosine residue. The two oxygen atoms of the nitro group present in compounds **C2** and **C5** showed hydrogen bonding with Glutamine, Asparagine, and Tyrosine amino acids.

Similarly, the oxygen atoms of methoxy group present in compound **C3** also participated in hydrogen bonding with Glycine, thereby increasing the affinity further. Substitution with hydroxyl group in compound **C4** also resulted in increased binding affinity as it participated in hydrogen bonding with Glutamine and Serine residues. Several hydrophobic interactions were also predicted between the chalcone scaffold and amino acid residues, such as Glycine, Tyrosine, Serine, Phenylalanine, Tryptophan, Leucine, and Isoleucine present at the binding pocket of the enzyme.

### 3.4. Molecular Dynamic (MD) Simulations

Usually, RMSD explains the conformational stability of the system, while RMSF provides an idea regarding residual flexibility. RMSD analysis deals with the determination of deviation of backbone atoms of protein during its initial to final conformation. Complex RMSD measures the scalar distance of backbone Cα atoms for protein as well as ligand during the whole simulation trajectory. The stability of the system is inversely related to RMSD [54]. As shown in Figure 9a, the AChE enzyme showed a deviation of 0.3 Å in RMSD during the first 10 ns, where it deviated from 0.6 to 0.9 Å. The AChE enzyme showed almost stable RMSD trajectory during the rest of the period with a deviation of ~0.07 Å. This system showed maximum RMSD of 0.9 Å at 10 ns. Whereas, the AChE-**C4** complex showed maximum deviation of almost ~0.6 Å during the first 13 ns, where it deviated from 1.58 to 1 Å. The AChE-**C4** complex showed almost stable trajectory with ~0.3 Å deviation until 60 ns. Maximum deviation of 1.58 Å was shown at 2 ns. As the AChE-**C4** complex showed RMSD deviation of less than 3 Å, it indicated the formation of a stable complex [55].

Similarly, RMSF explains residual flexibility and is inversely related to the stability of the system [56]. As shown in Figure 9b, only few significant residual fluctuations were seen for the free AChE enzyme with maximum RMSF of 1.25 Å. In comparison to the free enzyme, the AChE-**C4** complex showed some residual fluctuations but within optimal range. Maximum RMSF of 2.25 Å was observed for the AChE-**C4** complex system. As this value was again less than 3 Å, it signified that the AChE-**C4** complex was quite stable during situations, such as the aqueous environment. The periodic boundary condition for the apo-form of the enzyme AChE and the MD simulations of the AChE-**C4** complex is provided in Appendix A. The solvated and ionized forms of AChE enzyme-**C4** complex and AChE enzyme alone are provided as Appendix A

## 4. Conclusions

The promising AChE inhibitory activity shown by the chalcone derivatives prompted us to conduct mechanistic studies using in silico molecular docking and MD simulation techniques. Several basic chalcone derivatives with substitutions on the two phenyl rings were synthesized and their in vitro AChE inhibition assay was performed. All of the synthesized compounds showed moderate to good enzyme inhibition and strong binding with the AChE receptor, as shown by the binding energy results obtained using molecular docking. All of the functionalities present in the chalcone scaffold were involved in binding with the surrounding amino acid residues at the binding site by different intermolecular forces. Strong hydrogen bonding, π–π, π–cation, and hydrophobic interactions were present between the atoms present in the chalcone scaffold and the amino acids. The two lipophilic aryl rings, A and B, and their substitutions had considerable effects on binding with the receptors, which can further be exploited to prepare more effective chalcone derivatives. The findings of this study would pave the way to prepare more targeted modifications in the chalcone scaffold and to design derivatives that could show improved and selective binding and better AChE inhibition with lesser adverse effects.

## Figures and Tables

**Figure 1 molecules-27-03181-f001:**
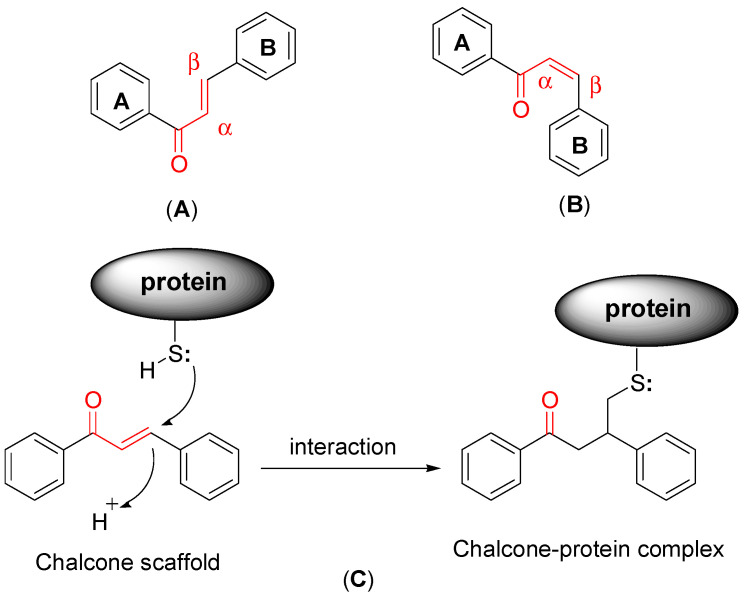
General structure of chalcone scaffold. (**A**) Trans (*E*) form; (**B**) cis (*Z*) form; (**C**) interaction of thiol/sulfhydryl group of cysteine residues of proteins with α, β-unsaturated carbonyl functionality of chalcones.

**Figure 2 molecules-27-03181-f002:**
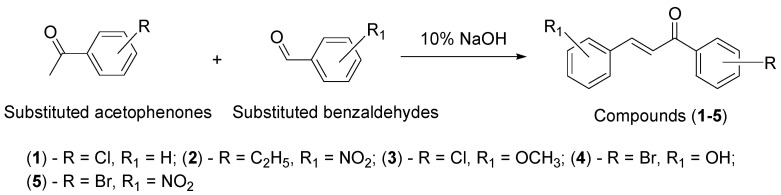
General route for the synthesis of titled compounds (**C1**–**C5**).

**Figure 3 molecules-27-03181-f003:**
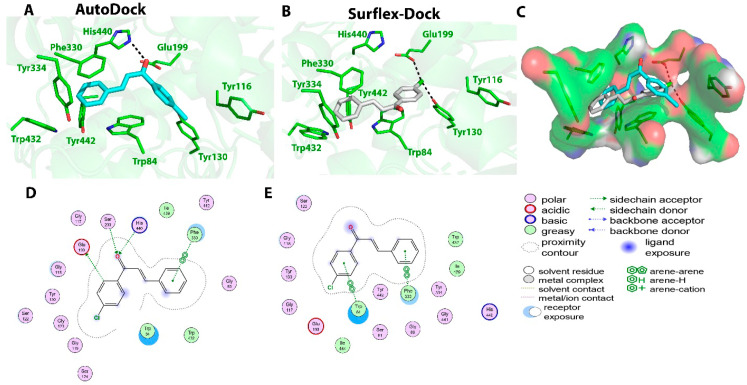
Biochemical interactions (**A**,**B**) show the 3D schematic diagrams of compound **C1** with AChE enzyme (PDB ID: 3I6M). Black dashed lines refer to the H-bonding interactions; amino acids implicated in H-bond with compound **C1** are shown as sticks. Diagrams were obtained using DSV software as docking results of AutoDock and Surflex-Dock programs, respectively; (**C**) shows the orientation of compound **C1** inside the AChE active-site gorge as predicted by AutoDock (blue) and Surflex-Dock (gray) and the electrostatic surface of the AChE gorge. Negative charges are shown in red color, positive charges in blue color, and neutral charges in white color; (**D**,**E**) are the 2D schematic diagrams obtained by Molecular Operating Environment software as docking results of AutoDock and Surflex-Dock, respectively representing the polar, acid, basic, and greasy amino acid around compound **C****1**.

**Figure 4 molecules-27-03181-f004:**
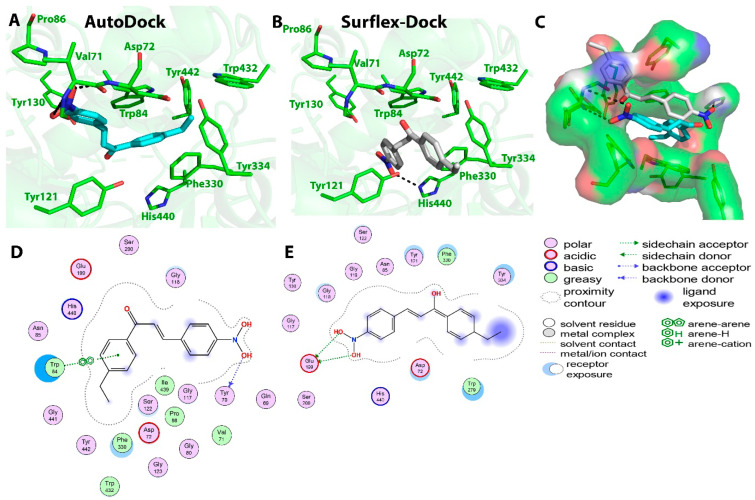
Biochemical interactions (**A**,**B**) show the 3D schematic diagrams of compound **C2** with AChE enzyme (PDB ID: 3I6M). Black dashed lines refer to the H-bonding interactions; amino acids implicated in H-bond with compound **C2** are shown as sticks. Diagrams were obtained using DSV software as docking results of AutoDock and Surflex-Dock programs, respectively; (**C**) shows the orientation of compound **C2** inside the AChE active-site gorge as predicted by AutoDock (blue) and Surflex-Dock (gray) and the electrostatic surface of the AChE gorge. Negative charges are shown in red color, positive charges in blue color, and neutral charges in white color; (**D**,**E**) are the 2D schematic diagrams obtained by Molecular Operating Environment software as docking results of AutoDock and Surflex-Dock, respectively representing the polar, acid, basic, and greasy amino acid around compound **C2**.

**Figure 5 molecules-27-03181-f005:**
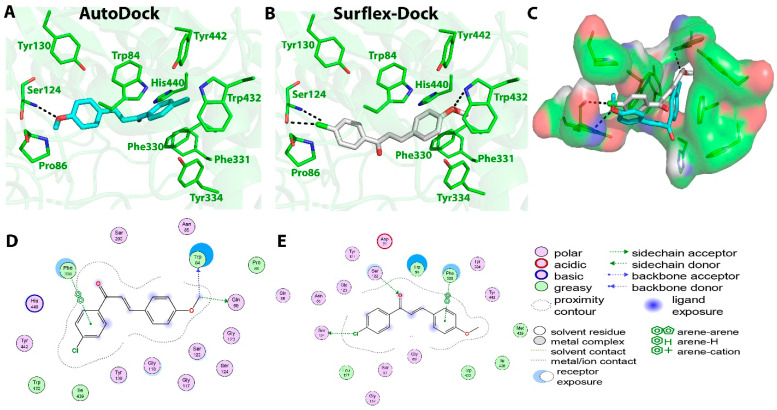
Biochemical interactions (**A**,**B**) show the 3D schematic diagrams of compound **C3** with AChE enzyme (PDB ID: 3I6M). Black dashed lines refer to the H-bonding interactions, amino acids implicated in H-bond with compound **C3** are shown as sticks. Diagrams were obtained using DSV software as docking results of AutoDock and Surflex-Dock programs, respectively; (**C**) shows the orientation of compound **C3** inside the AChE active-site gorge as predicted by AutoDock (blue) and Surflex-Dock (gray) and the electrostatic surface of the AChE gorge. Negative charges are shown in red color, positive charges in blue color, and neutral charges in white color; (**D**,**E**) are the 2D schematic diagrams obtained by Molecular Operating Environment software as docking results of AutoDock and Surflex-Dock, respectively representing the polar, acid, basic, and greasy amino acid around compound **C****3**.

**Figure 6 molecules-27-03181-f006:**
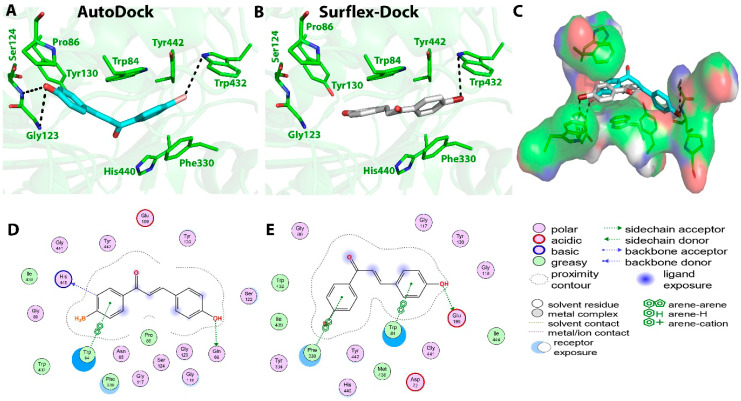
Biochemical interactions (**A**,**B**) show the 3D schematic diagrams of compound **C4** with AChE enzyme (PDB ID: 3I6M). Black dashed lines refer to the H-bonding interactions; amino acids implicated in H-bond with compound **C4** are shown as sticks. Diagrams were obtained using DSV software as docking results of AutoDock and Surflex-Dock programs, respectively; (**C**) shows the orientation of compound **C4** inside the AChE active-site gorge as predicted by AutoDock (blue) and Surflex-Dock (gray) and the electrostatic surface of the AChE gorge. Negative charges are shown in red color, positive charges in blue color, and neutral charges in white color; (**D**,**E**) are the 2D schematic diagrams obtained by Molecular Operating Environment software as docking results of AutoDock and Surflex-Dock, respectively representing the polar, acid, basic, and greasy amino acid around compound **C****4**.

**Figure 7 molecules-27-03181-f007:**
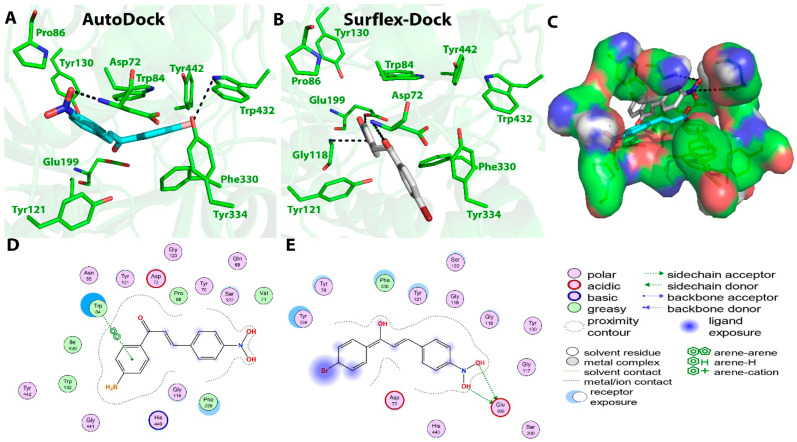
Biochemical interactions (**A**,**B**) show the 3D schematic diagrams of compound **C5** with AChE enzyme (PDB ID: 3I6M). Black dashed lines refer to the H-bonding interactions. Amino acids implicated in H-bond with compound **C5** are shown as sticks. Diagrams were obtained using DSV software as docking results of AutoDock and Surflex-Dock programs, respectively; (**C**) shows the orientation of compound **C5** inside the AChE active-site gorge as predicted by AutoDock (blue) and Surflex-Dock (gray) and the electrostatic surface of the AChE gorge. Negative charges are shown in red color, positive charges in blue color, and neutral charges in white color; (**D**,**E**) are the 2D schematic diagrams obtained by Molecular Operating Environment software as docking results of AutoDock and Surflex-Dock, respectively representing the polar, acid, basic, and greasy amino acid around compound **C****5**.

**Figure 8 molecules-27-03181-f008:**
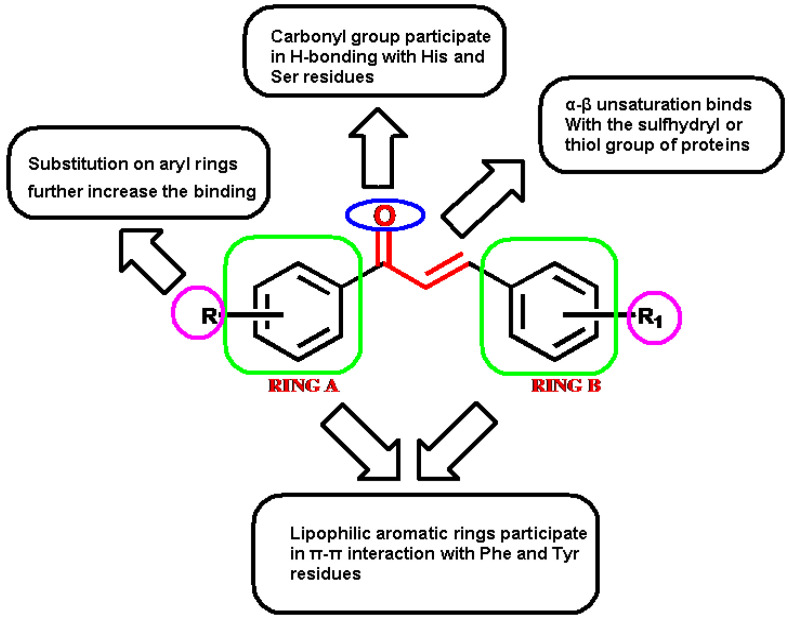
Functionalities of chalcone scaffold and their roles in binding to the AChE receptor.

**Figure 9 molecules-27-03181-f009:**
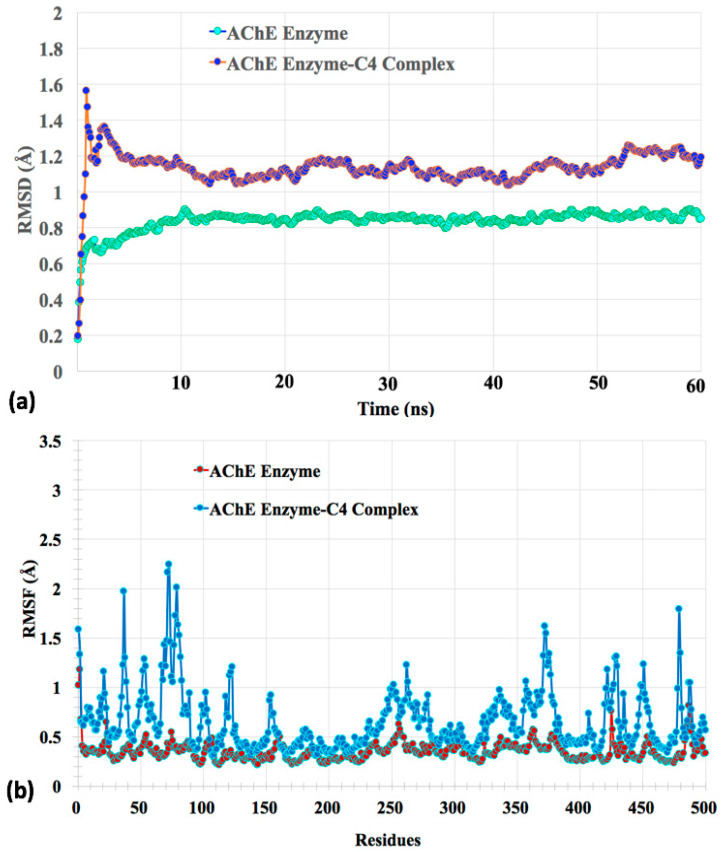
The MD calculated parameters. (**a**) The root mean square deviation (RMSD); (**b**) root mean square fluctuations (RMSF) for AChE enzyme and AChE enzyme-**C4** complex.

**Table 1 molecules-27-03181-t001:** AChE inhibitory activity (IC_50_ values) of synthesized chalcones and their binding energies calculated using AutoDock software.

Compound No.	Structure	Molecular Formula (Mol. Wt.)	CLogP Values	IC_50_ (μM)	Binding Energy (Kcal/mol)
**C1**	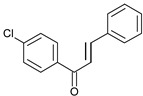	C_15_H_11_ClO (242.7)	4.65	22 ± 2.8	−8.4
**C2**	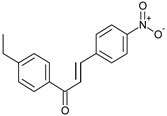	C_17_H_15_NO_3_ (281.3)	4.63	31.2 ± 1.8	−8.29
**C3**	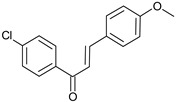	C_16_H_13_ClO_2_ (272.7)	4.57	32 ± 2.4	−8.14
**C4**	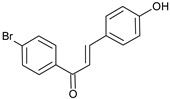	C_15_H_11_BrO_2_ (303.2)	4.13	36.9 ± 5.6	−8.55
**C5**	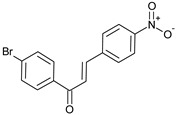	C_15_H_10_BrNO_3_ (331.0)	4.54	37.6 ± 0.75	−8.16

## Data Availability

The data associated with the study are available with the corresponding author and can be produced upon reasonable request.

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
