# Peer review of "Chalcone Scaffolds Exhibiting Acetylcholinesterase Enzyme Inhibition: Mechanistic and Computational Investigations"

_molecules, 2022, doi:10.3390/molecules27103181_

Round 1

Reviewer 1 Report

The Authors of manuscript described the ability of five synthetic chalcones to inhibition of acetylcholinesterase enzyme using mechanistic and computational investigations. It is interesting work but I have a few comments:

  1. Citation in whole manuscript is not in proper way, not according to Molecules template, also the style in heading 1 is not correct.
  2. Table 1 – please improve the structures because not all presented the compounds, which were described in manuscript in figure 1.
  3. Also Figure 8 is not complete – please improve it.
  4. There are no information about purity of synthesized compounds (using HPLC method or HRMS), lack of description of NMR spectra, yields of synthesis. Furthermore, chalcones in Table 1 are sometimes in cis or trans form (the double bond is up or down), which one is correct ? The compounds are known or novel ? Please explain.

Author Response

The Authors of manuscript described the ability of five synthetic chalcones to inhibition of acetylcholinesterase enzyme using mechanistic and computational investigations. It is interesting work but I have a few comments:

We thank the reviewer for his/her crucial comments. We addressed all the comments very carefully and the manuscript is revised accordingly.

  1. Citation in whole manuscript is not in proper way, not according to Molecules template, also the style in heading 1 is not correct.

Answer: We modified the manuscript and is formatted as per the journal’s instructions. The citation style as well as the heading style is modified. Also, the references are formatted as per the journal’s guidelines.

  1. Table 1 – please improve the structures because not all presented the compounds, which were described in manuscript in figure 1.

Answer: Thank you very much for the comment. We respectfully would like to mention that the structures in Table 1 are COMPLETE. The structure shown in Figure 1 is NOT the general structure of synthesized chalcones but is showing the binding of chalcones with the thiol/sulfhydryl group of proteins. The general structures of synthesized chalcones are shown in Figure 2 and Figure 8.   

  1. Also Figure 8 is not complete – please improve it.

Answer: Thank you again for the comment. Figure 8 is Correct and it should not be mistaken with Figure 1.

  1. There are no information about purity of synthesized compounds (using HPLC method or HRMS), lack of description of NMR spectra, yields of synthesis. Furthermore, chalcones in Table 1 are sometimes in cis or trans form (the double bond is up or down), which one is correct ? The compounds are known or novel ? Please explain.

Answer: Thank you very much for the crucial comment. Please note that the spectral data including the Mass (HRMS) and NMR data are included along with the obtained yield of each compound (Please see Page 4). The structures of chalcones in Table 1 are modified and uniformity in the structure is maintained as suggested (Please see Table 1 at page 6). Lastly, the synthesized compounds are known and were synthesized to study the extent and mechanism by which it binds to the acetylcholinesterase enzyme using molecular docking and molecular dynamics simulation and to establish the in vitro-in silico correlation.  

Reviewer 2 Report

This present work describes the design, synthesis, mechanistic investigations, molecular docking studies, and acetylcholinesterase inhibition profile of five chalcones with different substitutions. In general, the manuscript is well-written and provides ample evidence to support the claims. The aims and the biological questions of the present study are clear and well-articulated. The data are well-presented and the characterization of compounds is properly done. The discussion of the results is self-explanatory. The obtained results are important to better understand the inhibition of the AChE enzyme and to develop better AChE inhibitors. I recommend this article to publish in the “Molecules” journal; however, some important modifications are required nevertheless which are listed below:

1) Introduction part is too long. Some paragraphs are more appropriate for the discussion section

2) Mention the city and country of manufacturers for each instrument used

3) This manuscript is not well-cited. There are several related citations which have not been mentioned.

4) Why the authors used 2 different docking programs (Autodock and Surflex-Dock)? Is it necessary?

5) Why only five compounds were synthesized?

6) Authors should explain the abbreviations used at their first appearance in the manuscript (OE, OA, OD, OG?).

7) Manuscript is not formatted as per the journals’ instructions for authors.

8) References should be carefully checked and formatted according to the journal’s requirements

Author Response

This present work describes the design, synthesis, mechanistic investigations, molecular docking studies, and acetylcholinesterase inhibition profile of five chalcones with different substitutions. In general, the manuscript is well-written and provides ample evidence to support the claims. The aims and the biological questions of the present study are clear and well-articulated. The data are well-presented and the characterization of compounds is properly done. The discussion of the results is self-explanatory. The obtained results are important to better understand the inhibition of the AChE enzyme and to develop better AChE inhibitors. I recommend this article to publish in the “Molecules” journal; however, some important modifications are required nevertheless which are listed below:

We thank the reviewer for his/her encouraging words and important comments. All the comments made are addressed very carefully and the manuscript is revised accordingly.

1) Introduction part is too long. Some paragraphs are more appropriate for the discussion section

Answer: Accepted. The introduction part is shortened and few sentences are shifted to the discussion section as suggested (Please refer to Page 11, line 383-389).

2) Mention the city and country of manufacturers for each instrument used

Answer: Accepted. The city and country of the manufacturer of all instruments used in the study are provided (Page 3, line 109, 113, 114, 115).

3) This manuscript is not well-cited. There are several related citations which have not been mentioned.

Answer: Thank you very much for the important comment. We added seven recent relevant references to the manuscript. Please see the References no. 41-45 and 50-51.

4) Why the authors used 2 different docking programs (Autodock and Surflex-Dock)? Is it necessary?

Answer: We thank the reviewer for the crucial comment. We used two different docking programs to compare the results obtained through these programs. The 3D structures of chalcones were docked against the protein using academic docking tool AutoDock 4.2.2 and commercial docking tool Surflex-Dock 2.1. Two docking tools were selected to compare their results and to identify all the binding interactions that may be present between the chalcone scaffold and the amino acid residues present in the protein (Page 5, line 194-196; Page 12, line 407-410). 

5) Why only five compounds were synthesized?

Answer: The main aim of our study was to perform mechanistic studies over the chalcone scaffold and to identify the important functional groups which take part in binding to the acetylcholinesterase enzyme using molecular docking and molecular dynamics simulation techniques. Five basic chalcone moieties were synthesized having common electron withdrawing and releasing groups and the effect of these groups on binding to the receptor was studied using in silico methods.

6) Authors should explain the abbreviations used at their first appearance in the manuscript (OE, OA, OD, OG?).

Answer: OE stands for ε-Oxygen atom, OA = α-Oxygen atom; OD = δ-Oxygen atom; OG = γ-Oxygen atom. These were included at their first appearances. Please refer to Page 6 (line 256) and 7 (line 270, 272, 277).

7) Manuscript is not formatted as per the journals’ instructions for authors.

Answer: The whole manuscript is now formatted as per the journals’ instructions for authors.

8) References should be carefully checked and formatted according to the journal’s requirements

Answer: All the references are carefully checked and are formatted according to the journal’s requirements

Reviewer 3 Report

Regarding to the work with title "Chalcone scaffolds exhibiting acetylcholinesterase enzyme inhibition: Mechanistic and computational investigations” the following points were observed :

1-The Manuscript need extensive English revision by professional, as well as need major revision.

2-Similarity % was observed, I recommend to re-write the following lines again:45-47, 50, 71-76, 91-92, 111, 181-188

3- The number of evaluated compounds were not enough to build drug discovery story.

4- the synthesized compounds were not novel 

5- Just 1H-NMR and mass spectrums were observed but this is not enough for chemical synthesis, as well as no any data regarding these spectrums were written in the manuscript.   

Author Response

Regarding to the work with title "Chalcone scaffolds exhibiting acetylcholinesterase enzyme inhibition: Mechanistic and computational investigations” the following points were observed :

Answer: We thank the reviewer for his/her imperative comments. All the comments made by the reviewer are addressed carefully and subsequent changes in the manuscript were made using track changes.

1-The Manuscript need extensive English revision by professional, as well as need major revision.

Answer: Thank you very much for the comment. The whole manuscript was revised for English language mistakes and necessary corrections were made at all places.

2-Similarity % was observed, I recommend to re-write the following lines again:45-47, 50, 71-76, 91-92, 111, 181-188.

Answer: All the suggested sentences are re-written in order to improve the English language as well as to reduce the similarity. Please refer to their new location at line numbers 45-51; 68-73; 96-99; 196-203 and 383-389.

3- The number of evaluated compounds were not enough to build drug discovery story.

Answer: The main aim of our study was to perform mechanistic studies over the chalcone scaffold and to identify the important functional groups which take part in binding to the acetylcholinesterase enzyme using molecular docking and molecular dynamics simulation techniques. Five basic chalcone moieties were synthesized having commonly used electron withdrawing and releasing groups and the effect of these groups on binding to the receptor was studied using in silico methods.

4- the synthesized compounds were not novel 

Answer: Thank you again for your crucial comment. Yes, the compounds synthesized are not novel. The aim of our study was NOT to prepare novel chalcone derivatives but to perform the mechanistic investigations of chalcones showing anti-AChE activity. No such study is performed in the past using both experimental kinetics and theoretical molecular docking and molecular dynamics simulation methods. Identifying the role of functionalities present in the chalcone scaffold would help in designing novel chalcone analogs with improved and selective binding yielding more effective drugs with lesser side effects.

5- Just 1H-NMR and mass spectrums were observed but this is not enough for chemical synthesis, as well as no any data regarding these spectrums were written in the manuscript.   

Answer: We agree with the reviewer that 1H –NMR spectrum is not enough to characterize the synthesized compounds. Therefore, we included the MS and Elemental analysis data to the manuscript. Please refer to Page 4, line 130-155.

Round 2

Reviewer 3 Report

The work on the “Chalcone scaffolds exhibiting acetylcholinesterase enzyme in- 2 hibition: Mechanistic and computational investigations” was improved by the authors as requested in the last revision and a few points still need editing; it is recommended to add a paragraph regarding the other biological activities for chalcone derivatives like antiproliferative activities “Antiproliferative activities of some biologically important scaffolds” and https://doi.org/10.1016/j.ejmech.2017.02.002,  the English of the manuscript was improved and the similarity rate  was reduced accordingly, the manuscript seems well improved.

Author Response

The work on the “Chalcone scaffolds exhibiting acetylcholinesterase enzyme in- 2 hibition: Mechanistic and computational investigations” was improved by the authors as requested in the last revision and a few points still need editing; it is recommended to add a paragraph regarding the other biological activities for chalcone derivatives like antiproliferative activities “Antiproliferative activities of some biologically important scaffolds” and https://doi.org/10.1016/j.ejmech.2017.02.002,  the English of the manuscript was improved and the similarity rate  was reduced accordingly, the manuscript seems well improved.

Answer: We thank the reviewer for the encouraging words. As suggested, few sentences discussing antiproliferative activities of chalcones are added to the “Discussion” section (please refer to Page 11, lines 383-388). Also, two new references (no. 50 and 51) are added to the manuscript as suggested. Thank you very much again for the improvement.